# Adaptive Selection based Referring Image Segmentation

## ABSTRACT

Referring image segmentation (RIS) aims to segment a particular region based on a specific expression. Existing one-stage methods have explored various fusion strategies, yet they encounter two significant issues. Primarily, most methods rely on manually selected visual features from the visual encoder layers, lacking the flexibility to selectively focus on language-preferred visual features. Moreover, the direct fusion of word-level features into coarse aligned features disrupts the established vision-language alignment, resulting in suboptimal performance. In this paper, we introduce an innovative framework for RIS that seeks to overcome these challenges with adaptive alignment of vision and language features, termed the Adaptive Selection with Dual Alignment (ASDA). ASDA innovates in two aspects. Firstly, we design an Adaptive Feature Selection and Fusion (AFSF) module to dynamically select visual features focusing on different regions related to various descriptions. AFSF is equipped with scale-wise feature aggregator to provide hierarchically coarse features that preserve crucial low-level details and provide robust features for successor dual alignment. Secondly, a Word Guided Dual-Branch Aligner (WGDA) is leveraged to integrate coarse features with linguistic cues by word-guided attention, which effectively addresses the common issue of vision-language misalignment by ensuring that linguistic descriptors directly interact with masks prediction. This guides the model to focus on relevant image regions and make robust prediction. Extensive experimental results demonstrate that our ASDA framework surpasses state-of-the-art methods on RefCOCO, RefCOCO+ and G-Ref benchmark. The improvement not only underscores the superiority of ASDA in capturing fine-grained visual details but also its robustness and adaptability to diverse descriptions.

## CCS CONCEPTS

• **Computing methodologies** → **Scene understanding**; **Image segmentation**.

## KEYWORDS

referring image segmentation, vision-language alignment

## 1 INTRODUCTION

Referring image segmentation (RIS) [7, 15, 64] aims to predict a pixel-wise mask for objects referred to in a natural language expression. It yields great value for various applications such as language-based human-robot interaction [50] and image editing [2].

*ACM MM, 2024, Melbourne, Australia*
© 2024 Copyright held by the owner/author(s). Publication rights licensed to ACM.
ACM ISBN 978-x-xxxx-xxxx-x/YY/MM
https://doi.org/XXXXXXX.XXXXXXX

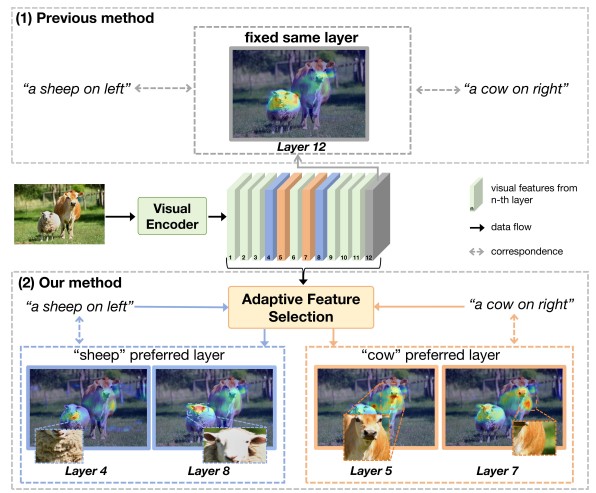

**(a) Paradigm comparisons**

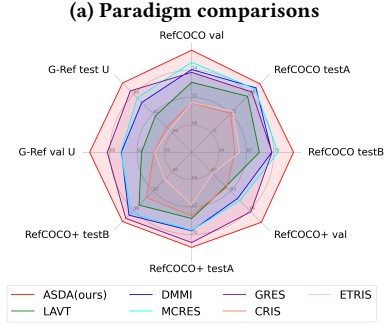

**(b) Qualitative comparisons**

**Figure 1:** (a) Comparison of feature selection paradigm: The previous method utilizes features from fixed layers within the visual encoder regardless of the linguistic content, while our method adaptively selects the language-preferred features based on the specific language input. (b) Qualitative comparisons reveal that our ASDA framework outperforms previous state-of-the-art methods, achieving the best results across all splits on three datasets.

Unlike standard semantic segmentation [10, 12, 13], which categorizes image pixels based on a fixed set of labels, RIS needs to understand free-form language expression to locate the exact pixels of the referenced object. Hence, the primary challenge of this task involves achieving precise alignment between the relevant visual content and the descriptive text at the pixel level, which is crucial for accurately producing the necessary mask.

Existing methods [15, 29, 40] often leverage external knowledge to facilitate learning, typically relying on separate vision and language encoder, such as the Swin visual encoder [35] paired with the BERT language encoder [24], which inherently lacks multi-modal correspondence. Meanwhile, some studies [51, 54] have built on well-aligned model like CLIP [42], leveraging the advantages of vision-language pretraining. Regardless of the backbone choice,

these methods commonly use manually determined visual features from the visual encoder layers, subsequently investing significant effort into feature fusion and alignment [4, 7, 18, 19, 23, 29, 34, 45, 51, 62]. Although these methods have promising performance, they encounter inherent challenges. Primarily, most methods rely on manually selected visual features from the visual encoder layers. As illustrated in Figure 1, when processing the same image with varying referring texts like "a sheep on the left" and "a cow on the right", these methods extract the same manually determined visual features. This manner lacks the flexibility to selectively focus on visual features that are more relevant to the specific language used. Moreover, the direct fusion between irrelevant visual features [7, 17, 51] with sentence-level language features introduces noise to subsequent word-level language features guided mask prediction, which results in disrupting the established vision-language alignments. As depicted in Figure 4, by visualizing the coarse features and the fine features, we observe that the fine features becomes disrupted after the interaction between the coarse features and word-level language features, focusing on regions unrelated to the textual description.

To this end, we propose an Adaptive Selection with Dual Alignment (ASDA) framework to enhance performance on RIS tasks, which mainly contains Adaptive Feature Selection and Fusion (AFSF) module and Word Guided Dual-Branch Aligner (WGDA) module. The Adaptive Feature Selection and Fusion (AFSF) module composes of Adaptive Feature Selection (AFS) module and Scale-Wise Feature Aggregator (SFA) module. Initially, AFS leverages sentence-level language features to select the most relevant features from the visual encoder layers. As shown in Figure 1, our AFS module can adaptively choose features that are more relevant to object described in the referring expression. Subsequently, SFA merges multi-scale visual features from AFS with sentence-level language features, effectively integrating semantics with the visual attributes of various layers. This provides hierarchically aggregated features that preserve crucial low-level details and offer robust features for successor dual alignment. Furthermore, we design two distinct branches in WGDA: the Coarse-to-Fine Segmentation Decoder (CFS) and the Word Guided Coefficient Generator (WCG), to more effectively interact word-level features with coarse features. The CFS utilizes robust features from AFSF module to generate candidate masks. Meanwhile, the WCG module employs a word-level language feature guided attention mechanism to generate the coefficients to combinate these masks. Unlike previous methods [22, 51, 59] that directly merge word-level language features into coarse features, our WGDA enables linguistic descriptors to directly interact with mask prediction without compromising the already aligned features, which helps get finely aligned visual feature.

In summary, the contribution of this work is fourfold:

- We propose an Adaptive Selection with Dual Alignment (ASDA) framework to enhance performance on RIS task.
- Our Adaptive Feature Selection and Fusion (AFSF) module dynamically selects the most relevant visual features based on language features, moving away from the conventional fixed-layer selection to a dynamic, text-responsive feature selection mechanism.

- Our Word Guided Dual-Branch Aligner (WGDA) module improves alignment and robustness of word-level features with visual features through a dual-branch structure, enhancing the interaction between linguistic descriptors and mask predictions.
- Our ASDA achieves state-of-the-art results on the RefCOCO, RefCOCO+, and RefCOCOg datasets.

## 2 RELATED WORK

**Referring Image Segmentation (RIS)** is designed to localize objects within images guided by natural language descriptions. The early approach [15] utilized a fusion technique combining linguistic and visual elements through concatenation. Subsequent efforts [4, 17, 21, 22, 29, 38] have harnessed sentence-level textual features from the descriptive phrases, whereas other studies [1, 9, 34, 40] have adopted word-level textual features for textual representation. Given that natural language intrinsically contains structured data [44, 57] that can be exploited to align with visual features, certain methodologies have decomposed expressions into various components [19, 52, 56, 58] or implemented a soft division approach using attention mechanisms [7, 10, 18, 45, 55, 62, 64].

Recent work has adopted more efficient structures for vision-language fusion. LAVT [59] utilizes the Swin Transformer [35] for visual tasks and incorporates modules for vision-language integration in the last four layers of the visual encoder. In contrast, ReSTR [25] and CRIS [51] start by separately encoding visual and linguistic inputs with a dual encoder, then merging these features either through a multi-modal transformer encoder or a cross-modal decoder. Inspired by the advancements in large language models [43, 48], new studies approach RIS as an auto-regressive generation task, introducing capabilities for logical reasoning [26, 63]. Recently, VPD [65] explores using semantic data from diffusion models [14, 46] for RIS applications, while ReLA [33] and DMMI [16] extend RIS capabilities to handle multiple targets.

Despite significant advances in RIS architecture, most methods face two main limitations. First, they depend on manually selected visual features from encoder layers, lacking flexibility to focus on language-preferred visual features for different objects. Additionally, most methods directly fuse word-level features with coarse visual features, leading to suboptimal performance. We introduce an innovative framework for RIS that seeks to overcome these challenges with adaptive selection and dual alignment of vision and language features.

**Vision-Language Model (VLM)** is a type of deep learning model designed to simultaneously interpret visual and textual information. These models can be categorized into two workflows: single-stream and dual-stream. Single-stream models, such as [3, 5, 6, 36], integrate vision and language embeddings using a unified self-attention encoder. In contrast, dual-stream models like CLIP [51], ALIGN [20], FILIP [61], and GLIP [28], employ separate encoders for each modality, aligning the outputs through a dot product. Other models such as ViLBERT [36] and LXMERT [47] utilize dual self-attention-based encoders to process within-modality interactions, while cross-attention mechanisms are used to handle interactions between modalities. As a milestone, CLIP [39] applies a contrastive

learning approach across a vast dataset of image-text pairs, demonstrating significant transfer capabilities across over 30 classification datasets. Following this model, CRIS [51] uses a transformer decoder in conjunction with a CLIP model, adapting CLIP's text-to-image matching expertise to text-to-pixel applications. Given the outstanding performance of CLIP [39], we follow this well-aligned vision-language model to implement our framework.

## 3 METHOD

### 3.1 Overview

The framework of our proposed Adaptive Selection with Dual Alignment (ASDA) is illustrated in Figure 2. An image $I$ and referring expressions $T$ are fed to ViT-B based CLIP visual encoder (12 layers) and language encoder [39] respectively to extract layer-wise visual features $F_V^{(i)}$ and word-level, sentence-level language features $[f_T, f_E]$ (see Section 3.2). Next, Adaptive Feature Selection (AFS) module adaptively selects low-level visual features $F_L$ from layers 4-6, middle-level visual features $F_M$ from layers 7-9, high-level visual features $F_H$ from highest layer of the Vision Encoder and generates multi-scale features $\tilde{F}_L, \tilde{F}_M$. Our Scale-Wise Feature Aggregator (SFA) then progressively fuses the global language features $f_E$ with multi-scale visual features $\tilde{F}_L, \tilde{F}_M, F_H$, generating coarse aligned feature $F_{coarse}$ (see Section 3.3). Finally, the Coarse-to-Fine Segmentation Decoder (CFS) processes $F_{coarse}$ through Local Visual Attention and segmentation decoder to generate response masks $M'$. The Word-Guided Coefficient Generator (WCG) first combines the attention-enhanced feature $F_A$ with word-level language feature $f_T$ using Visual and Language Local Attention, generating fine feature $F_{fine}$. Following this, the Gated Coefficient Generation produces the coefficients $f_{coef}$ corresponding to the masks $M'$. The final mask output $M \in \mathbb{R}^{H \times W}$ is then obtained by applying a weighted sum operation between the masks $M'$ and the coefficients $f_{coef}$ (see Section 3.4).

### 3.2 Features Extraction

Given an image $I$ and referring expressions $T$, we extract layer-wise visual features $F_V^{(i)}$ and text features $[f_T, f_E]$ through ViT-B-based CLIP visual encoder and language encoder, respectively.

**Image encoder.** Following the design of vision transformer ViT, the image $I \in \mathbb{R}^{H_I \times W_I \times 3}$ is patched and projected to $I_P \in \mathbb{R}^{H \times W \times C}$, where $(H, W) = (H_I/P, W_I/P)$ and $P$ indicates the resolution of each image patch. Then, $I_P$ is fed into ViT [8] which employs $N$ transformer layers. And the output of layer $i$ is defined as $F_V^{(i)} \in \mathbb{R}^{H \times W \times C}$, $i = 0, \ldots, N$. The visual features of the final layer are recorded as the highest-level visual features $F_H$. Especially, there are 12 transformer layers in ViT-B. Each layer's visual features contain a learnable embedding which is called class token. The class token is recorded as the global visual features $f_C^{(i)} \in \mathbb{R}^C$.

**Text encoder.** For the given natural language referring expression, we extract word-level language feature $f_T \in \mathbb{R}^{N_T \times C}$ using a modified Transformer [49] architecture described in CLIP. Here, $N_T$ represents the number of word-level language tokens. The expression sequence is enclosed with [SOS] and [EOS] tokens to indicate the start and end of sequences. The activations of the [EOS] token

are considered as the sentence-level language feature $f_E \in \mathbb{R}^C$ for the entire natural language expression.

### 3.3 Adaptive Feature Selection and Fusion

As shown in Figure 2, the Adaptive Feature Selection and Fusion (AFSF) module consists of two main components: Adaptive Feature Selection (AFS) and Scale-Wise Feature Aggregator (SFA). Specifically, we design an adaptive selection network in ASF which is used to selectively identify the most relevant low-level visual feature $F_L$ and mid-level visual feature $F_M$. Then it generates corresponding multi-scale features of low-level visual feature and mid-level visual feature, denoted as $\tilde{F}_L$ and $\tilde{F}_M$. Subsequently, the SFA module establishes the relationship between global language feature $f_E$ and the highest-level local visual features $F_H$. Additionally, it fuses multi-scale features of low-level visual feature $\tilde{F}_L$ and mid-level visual feature $\tilde{F}_M$ to enhance the visual information of the target object, generating coarse feature $F_{coarse}$.

**Adaptive Feature Selection (AFS).** Based on our visual analysis in Figure 1, the visual features in different layers of the visual encoder can capture different local highlights related to various objects. When selecting the low-level visual feature $F_L$, the input includes global visual feature of the low-level visual feature candidate layers $f_C^l$, where $l \in \{4, 5, 6\}$, and the sentence-level language feature $f_E$. We first compute the cosine similarity between $f_E$ and the class token $f_C^l$ for each $l \in \{4, 5, 6\}$:

$$score^l = f_E \otimes \Phi_T(f_C^l), \quad l \in \{4, 5, 6\} \tag{1}$$

where $score^l \in \mathbb{R}^C$, $\otimes$ means element-wise multiplication, $\Phi_T$ is a linear layer that maps the global visual feature $f_C^l$ to the same dimension as $f_E \in \mathbb{R}^{1 \times C}$. The score tokens $score^l$ serve a pivotal role in determining the alignment and relevance between textual descriptions and visual representations at different levels. They measure the similarity of visual features to textual descriptions, illustrating the relevance degree of correspondence. We then design an adaptive selection network based on $score^l$ to select the low-level visual feature:

$$\begin{aligned} score_L &= \Phi_{as}([score^4, socre^5, score^6]) \\ L &= \Phi_{argmax}(score_L) \\ F_L &= F_V^{(L)} \end{aligned} \tag{2}$$

where $[,]$ denotes concatenation, $\Phi_{as}$ means adaptive selection network which is a combination of a Linear layer and softmax, $score_L \in \mathbb{R}^3$ represents the relevance scores of different feature layers processed through $\Phi_{as}$, $L$ is the index of the most relevant feature layer identified by $\Phi_{as}$ using $\Phi_{argmax}$, and $F_L$ is the feature of the ViT layer corresponding to the index $L$. $\Phi_{as}$ here serves a critical function in identifying the most appropriate feature layer index from the concated score tokens. There are many possible network architectures for $\Phi_{as}$, but our experiments show that a simple combination of a linear layer and softmax yields the best results, as detailed in Table 5. The extraction of mid-level visual features $F_M$ follows the similar way applied for low-level features, employing our proposed Adaptive Feature Selection network $\Phi_{as}$ to select the most related feature. This process is encapsulated as

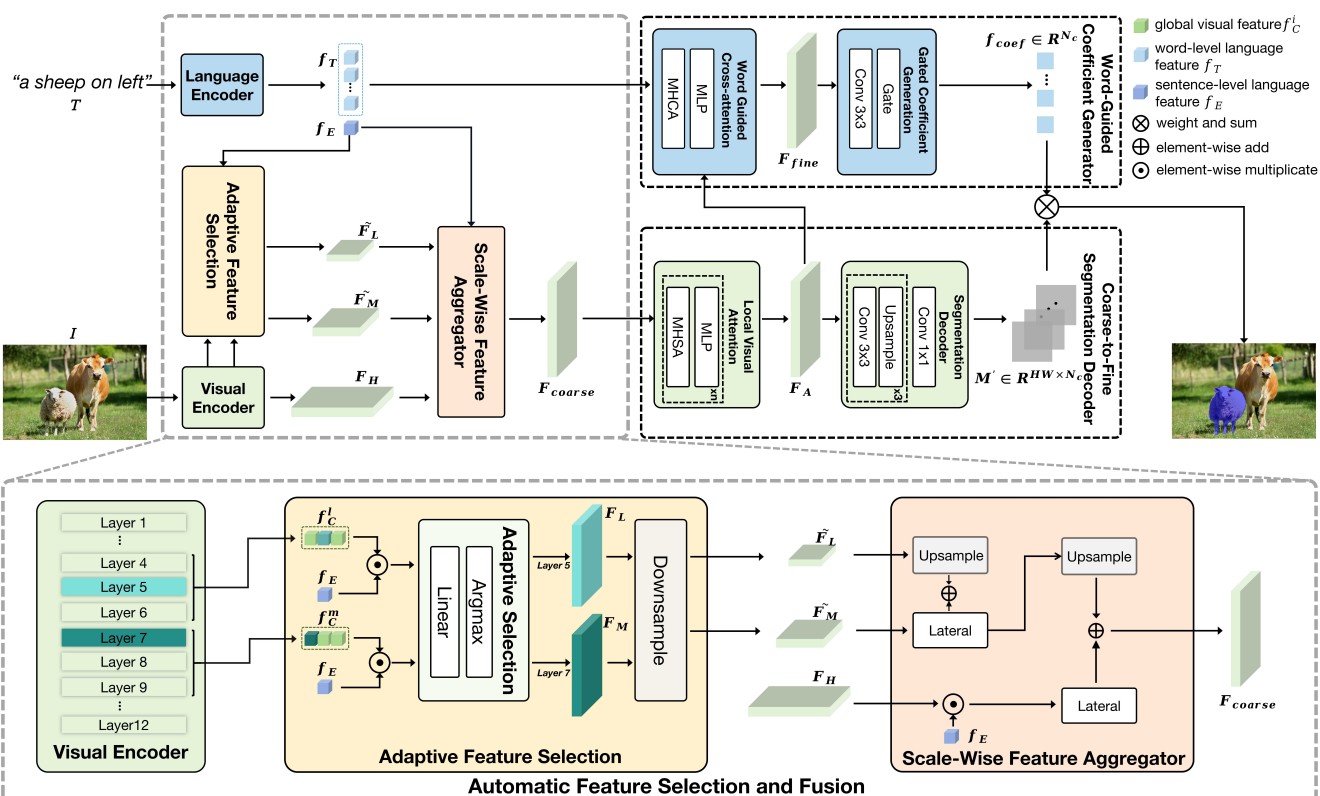

**Figure 2:** The overview of ASDA framework. It begins with extracting visual and language features from image $I$ and referring text $T$. The Adaptive Feature Selection (AFS) module adaptively selects visual features $F_L$ from Layers 4-6 and $F_M$ from Layers 7-9 of the Visual Encoder and generates multi-scale features $\tilde{F}_L$, $\tilde{F}_M$. Our Scale-Wise Feature Aggregator (SFA) then progressively fuses the global language features $f_E$ with high-level feature $F_H$ together with multi-scale visual features $\tilde{F}_L$ and $\tilde{F}_M$, generating coarse feature $F_{coarse}$. Finally, the Coarse-to-Fine Segmentation Decoder (CFS) and Word-Guided Coefficient Generator (WCG) generate the final mask by using response masks $M'$ and coefficients $f_{coef}$.

follows:

$$score^m = f_E \cdot \Phi_T(f_C^m), \quad m \in \{7, 8, 9\}$$
$$M = \Phi_{argmax}(\Phi_{as}([score^7, score^8, score^9])) \quad (3)$$
$$F_M = F_V^{(M)}$$

Thus, we obtain the low-level visual features $F_L$, mid-level visual features $F_M$. Here we consider the output of the last layer of ViT-B as the highest-level feature $F_H$, which captures more global information in the image. All these three features have the same dimension $\mathbb{R}^{H \times W \times C}$. We then downsample the low-level and mid-level features. This process yields $\tilde{F}_M \in \mathbb{R}^{\frac{H}{2} \times \frac{W}{2} \times C}$ for mid-level feature and $\tilde{F}_L \in \mathbb{R}^{\frac{H}{4} \times \frac{W}{4} \times C}$ for low-level feature, facilitating a more detailed and hierarchical representation of visual information.

**Scale-Wise Feature Aggregator (SFA).** Inspired by the effectiveness of multi-scale features in the visual domain [11, 31, 67], we have adopted multi-scale features to enhance feature alignment. Unlike traditional tasks in the visual domain such as object detection [11, 31, 67] and segmentation [27, 66], maintaining the positional relationship between text and images when using multi-scale features in RIS settings presents a challenge. To solve this

problem, we introduce the Scale-Wise Feature Aggregator (SFA) to preserve the spatial relationship between text and images from CLIP while leveraging multi-scale features. SFA initially combines global language feature $f_E$ with top-layer visual features $F_H$, using element-wise multiplication:

$$f_G = f_H \odot f_E \quad (4)$$

where $f_H \in \mathbb{R}^C$ denotes the element of the top-layer visual features $F_H$ and $f_G \in \mathbb{R}^C$ represents the single element of the global-to-local fused feature $F_G$. Subsequently, the global-to-local fused feature $F_G \in \mathbb{R}^{H \times W \times C}$ is integrated with the down-sampled low-level visual features $\tilde{F}_L \in \mathbb{R}^{\frac{H}{4} \times \frac{W}{4} \times C}$ and mid-level visual features $\tilde{F}_M \in \mathbb{R}^{\frac{H}{2} \times \frac{W}{2} \times C}$ through the following gradual process:

$$F'_L = \Phi_{up}(\tilde{F}_L)$$
$$F'_M = \Phi_{lateral}(\tilde{F}_M) + F'_L \quad (5)$$
$$F_{coarse} = \Phi_{agg}(\Phi_{lateral}(F_G) + \Phi_{up}(F'_M))$$

where the down-sampled low-level visual feature $\tilde{F}_L$ is initially 2x up-sampled by $\Phi_{up}$ to $F'_L \in \mathbb{R}^{\frac{H}{2} \times \frac{W}{2} \times 32}$ with channel reduction to 32. Concurrently, the down-sampled mid-level visual feature

$\tilde{F_M}$ is enhanced through $\Phi_{\text{lateral}}$ to achieve $F'_M \in \mathbb{R}^{H \times W \times 32}$, with channel reduction to 32. $\Phi_{\text{lateral}}$ here is a $1 \times 1$ Convolution-ReLU block. Based on our assumption that only basic information about the object location is needed, we boldly reduce the channel dimension to lower computational demands. The final aggregation step combines these enhanced features into $F_{\text{coarse}} \in \mathbb{R}^{HW \times C}$. This is achieved by employing a $3 \times 3$ convolution followed by a flattening operation via $\Phi_{\text{agg}}$. This process is designed to restore the channel dimensions and then flatten the output for subsequent stages, effectively maintaining a balance between preserving details and ensuring computational efficiency in our approach to integrating multi-scale features.

### 3.4 Word Guided Dual-Branch Aligner

As shown in Figure 4, the coarse feature map of CRIS [51] become disrupted after interaction with word-level language features, often focusing on regions unrelated to textual description. The previous single-branch alignment method [25, 51, 54] utilizes a Transformer Encoder or Decoder architecture to fuse coarse features with word-level language features, as demonstrated in Figure 3. To address this issue, we develop two distinct branches within the WGDA: Coarse-to-Fine Segmentation Decoder (CFS) and Word Guided Coefficient Generator (WCG), which more effectively align word-level features with coarse feature $F_{coarse}$. Specifically, the CFS branch utilizes Local Visual Attention and Segmentation Decoder to discern relationships among visual tokens and generate masks $M' \in \mathbb{R}^{HW \times N_c}$ focusing on different parts. The WCG branch comprises Visual and Language Local Attention and Gated Coefficient Generation, which find correspondences between word-level language features and local visual features to generate coefficients $f_{coef} \in \mathbb{R}^{N_c}$ for different response masks $M'$. And then we can get the final mask output $M \in \mathbb{R}^{H \times W}$ by applying a weighted sum operation between masks $M'$ and coefficients $f_{coef}$. This design effectively preserves the alignment between global language and visual features while seamlessly incorporating word-level language features, avoiding the issues that come with single-branch.

**Coarse-to-Fine Segmentation Decoder (CFS).** Within the CFS branch, we first leverage the coarse aligned feature $F_{coarse}$ as inputs to Local Visual Attention module, resulting in the attention-enhanced local visual features $F_A \in \mathbb{R}^{HW \times C}$. For Local Visual Attention module with $n$ layers, the workflow of i-th layer is simplified as follows:

$$F_a^{(i-1)} = \Phi_{\text{MHSA}}(\Phi_{\text{LN}}(F_{coarse})) + F_{coarse}$$
$$F_a^{(i)} = \Phi_{\text{MLP}}(F_a^{(i-1)}) + F_a^{(i-1)}. \quad i = 1, 2, ..., n \quad (6)$$
$$F_A = F_a^{(n)}$$

where $F_a^{(i-1)}$ represents the gradual refined visual features, $\Phi_{\text{MHSA}}$ indicates a multi-head self-attention layer and $\Phi_{\text{LN}}$ means Layer Normalization. After $n$ layers of self-attention interaction, the Segmentation Decoder leverages the attention-enhanced visual features $F_A$ to produce the response masks $M'$, which is computed as

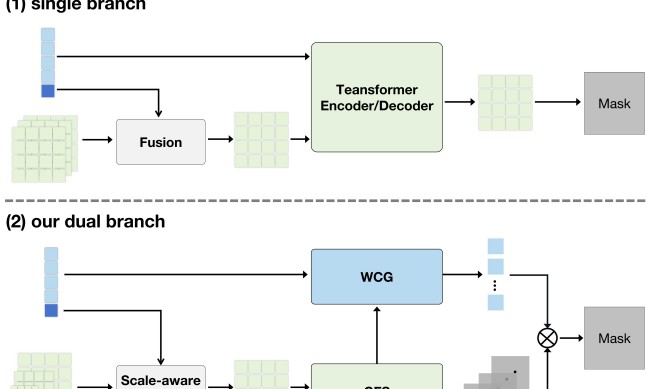

**Figure 3:** Illustration of single-branch cross-modal alignment in existing RIS methods and our dual alignment that enables linguistic descriptors to directly interact with mask prediction.

follows:

$$F_A^{(i)} = \Phi_{\text{conv3}\times3}(\Phi_{\text{up}}(F_A^{i-1})), \quad i = 1, 2, 3$$
$$F_A^{(4)} = \Phi_{\text{conv1}\times1}(F_A^{(3)}) \quad (7)$$
$$M = \Phi_{\text{sigmoid}}(F_A^{(4)})$$

where $\Phi_{\text{up}}$ represents the upsampling operation, $\Phi_{\text{conv3}\times3}$ denotes the convolution operation with a $3 \times 3$ kernel, $F_A^{(i)}$ denotes the feature after the $i$-th convolution and upsampling operation, and $\Phi_{\text{conv1}\times1}$ indicates the final convolution operation with a $1 \times 1$ kernel. After the last convolution, the $\Phi_{\text{sigmoid}}$ function is applied to $F_A^{(4)}$ and then produce the response masks $M' \in \mathbb{R}^{HW \times N_c}$. Note that the hyperparameter n and $N_c$ is discussed in the Table 3 and 4 in ablation study.

**Word-Guided Coefficient Generator (WCG).** Within WCG branch, we integrate the refined visual features $F_A$ and the word-level language features $f_T$ through the Visual and Language Local Attention module. The multi-head cross-attention layer is adopted to propagate fine-grained semantic information into the evolved visual features. The calculation is as follows:

$$F'_K = \Phi_{\text{MHCA}}(\Phi_{\text{LN}}(F_A, f_T)) + F_A$$
$$F_{fine} = \Phi_{\text{MLP}}(F'_K) + F'_K \quad (8)$$

where $\Phi_{\text{MHCA}}$ denotes the multi-head cross-attention layer, and $F'_K$ is the intermediate features. The evolved multi-modal fine feature $F_{fine} \in \mathbb{R}^{HW \times C}$ which captures the relationship between word-level language features and local visual features is utilized for generating the coefficient $f_{coef}$ through Gated Coefficient Generation module. The calculation is as follows:

$$f_{coef} = \Phi_{\text{coef}}(F_{fine}) \quad (9)$$

where $\Phi_{\text{coef}}$ comprises two stacked $3 \times 3$ convolution layers and one $1 \times 1$ convolution layer, followed by the Tanh activation function. The WCG branch effectively captures the essence of visual-textual interplay, producing normalized coefficients $f_{coef} \in \mathbb{R}^{N_c}$. Coefficients $f_{coef}$ are then used to guide the segmentation output, ensuring a coherent integration of cross-modal insights.

**Table 1:** Comparisons with the state-of-the-art approaches on three benchmarks. We report the results of our method with various visual backbones. U: The UMD partition. G: The Google partition. "-" represents that the result is not provided. IoU is utilized as the metric.

| Method | Visual | Language | RefCOCO | | | RefCOCO+ | | | G-Ref | | | Avg |
| | | | val | testA | testB | val | testA | testB | val(U) | test(U) | val(G) | |
|---|---|---|---|---|---|---|---|---|---|---|---|---|
| CGAN [37] | DarkNet-53 | Bi-GRU | 64.86 | 68.04 | 62.07 | 51.03 | 55.51 | 44.06 | 51.01 | 51.69 | 46.54 | 54.98 |
| LTS [22] | DarkNet-53 | Bi-GRU | 65.43 | 67.76 | 63.08 | 54.21 | 58.32 | 48.02 | 54.40 | 54.25 | - | 58.18 |
| ReSTR [25] | ViT-B-16 | GloVe | 67.22 | 69.30 | 64.45 | 55.78 | 60.44 | 48.27 | 54.48 | - | - | 59.99 |
| LAVT [59] | Swin-B | BERT | 72.73 | 75.82 | 68.79 | 62.14 | 68.38 | 55.10 | 61.24 | 62.09 | 60.50 | 65.20 |
| VLT [7] | Swin-B | BERT | 72.96 | 75.96 | 69.60 | 63.53 | 68.43 | 56.92 | 63.49 | 66.22 | 62.80 | 66.66 |
| SLViT [41] | SegNeXt | BERT | 74.02 | 76.91 | 70.62 | 64.07 | 69.28 | 56.14 | 62.75 | 63.57 | 60.94 | 66.48 |
| SADLR [60] | Swin-B | BERT | 74.24 | 76.25 | 70.06 | 64.28 | 69.09 | 55.19 | 63.60 | 63.56 | 61.16 | 66.38 |
| DMMI [16] | Swin-B | BERT | 74.13 | 77.13 | 70.16 | 63.98 | 69.73 | 57.03 | 63.46 | 64.19 | 61.98 | 66.87 |
| MCRES [53] | Swin-B | BERT | 74.92 | 76.98 | 70.84 | 64.32 | 69.68 | 56.64 | 63.51 | 64.9 | 61.63 | 67.05 |
| ReLA [33] | Swin-B | BERT | 73.82 | 76.48 | 70.18 | 66.04 | 71.02 | 57.65 | 65.00 | 65.97 | 62.7 | 67.65 |
| CRIS [51] | CLIP-R101 | CLIP | 70.47 | 73.18 | 66.1 | 62.27 | 68.08 | 53.68 | 59.87 | 60.36 | - | 64.25 |
| ETRIS [54] | CLIP-ViT-B | CLIP | 70.51 | 73.51 | 66.63 | 60.10 | 66.89 | 50.17 | 59.82 | 59.91 | 57.88 | 62.82 |
| **ASDA** | **CLIP-ViT-B** | **CLIP** | **75.06** | **77.14** | **71.36** | **66.84** | **71.13** | **57.83** | **65.73** | **66.45** | **63.55** | **68.34** |

Finally, the segmentation mask $M$ is obtained by weighting and summing the coefficients $f_{coef} \in \mathbb{R}^{N_c}$ and the response mask $M' \in \mathbb{R}^{HW \times N_c}$:

$$M = \Phi_{\text{reshape}}(f_{coef} \otimes M') \qquad (10)$$

where $M \in \mathbb{R}^{H \times W}$ represents the final output mask, $\otimes$ denotes the element-wise multiplication, and $\Phi_{\text{reshape}}$ is a reshaping operation that transforms the multiplied result into the desired output dimensions. Rather than adopting the contrastive learning proposed in CRIS, we supervise mask prediction through a linear combination of focal loss [32] and dice loss [30].

## 4 EXPERIMENTS

### 4.1 Datasets

We conduct extensive experiments on three benchmark datasets. **RefCOCO** stands out in this research domain, comprises 19, 994 images and 142, 210 referring expressions linked to 50, 000 objects. The dataset is divided into 120, 624 training, 10, 834 validation, 5, 657 test A, and 5, 095 test B images. It is characterized by typically containing two or more objects per image, with referring expressions averaging 3.6 words in length. **RefCOCO+** introduces an elevated challenge by omitting expressions containing certain absolute-location words. It includes 19, 992 images, presenting 49, 856 objects through 141, 564 linguistic expressions. The dataset is distributed across 120, 624 training, 10, 758 validation, 5, 726 test A, and 4, 889 test B samples. **G-Ref** distinguishes itself by using Amazon Mechanical Turk to collect 104, 560 referring expressions that describe 54, 822 objects in 26, 711 images. This collection method ensures greater linguistic diversity in the expressions, which averages 8.4 words and frequently mention locations and appearances. G-Ref is available in two versions, curated by the University of Maryland (UMD) and Google. Both versions have been utilized in our experiment.

### 4.2 Implementation Details

**Experimental Settings.** We firstly use ViT-B as the visual encoder and Transformer as the language encoder. Both the visual and language encoders are initialized with CLIP. Input images are resized to $416 \times 416$. The number $N_c$ of response masks $M'$ and coefficients $f_{coef}$ are set to 32. The parameter $n$ of Local Visual Attention module layers in CFS module is set to 2. The maximum length for the input natural language expression is set to 17 for RefCOCO and RefCOCO+, and 22 for G-Ref, including the [SOS] and [EOS] tokens. We use the Adam optimizer to train the network for 30 epochs with an initial learning rate of $5 \times 10^{-5}$, and we decay the learning rate in the $18^{th}$, $25^{th}$ epochs with a decay rate of 0.1. We train the model with a batch size of 28 on 2 RTX 3090 GPUs.

**Metrics.** Following previous works, we adopt two metrics to verify the effectiveness: overall Intersection over Union (IoU) and Precision@$X$. IoU calculates the ratio of intersection to union regions between the predicted segmentation mask and the ground truth. Precision@$X$ measures the percentage of test images that achieve an IoU score exceeding the threshold $X$, with $X$ values of 0.5, 0.7, and 0.9.

### 4.3 Comparison with State-of-the-Art Methods

We evaluate our ASDA against state-of-the-art methods that utilize various visual and language backbones. Table 1 shows performance comparisons on three common splits of RefCOCO, RefCOCO+ and G-Ref. Compared to the recently SOTA ReLA [33] which uses Swin-B visual backbone [35] and BERT language backbone [24], our ASDA improves IoU by 1.24%, 0.66% and 1.18% on val, testA and testB splits of RefCOCO, respectively. On RefCOCO+, ASDA shows improvements of 0.80%, 0.11%, and 0.18% respectively for val, testA, and testB splits. Furthermore, on G-Ref dataset, improvements are 0.73% for val (U), 0.48% for test (U), and 0.85% for val (G) splits. This demonstrates that our ASDA not only achieves a better understanding of the location and appearance information in RefCOCO but also adapts to the various forms of expressions in RefCOCO+ and G-Ref. Besides, the following two comparisons show the effectiveness of our ASDA from different perspectives: (1) CRIS [51] is the first method which uses CLIP-R101 in RIS task. Compared to CRIS, our ASDA demonstrates significant performance improvements, with average gains of 4.6%, 3.92%, and 5.13% on RefCOCO, RefCOCO+, and G-Ref datasets, respectively. This demonstrates

**Table 2:** Ablation study on the validation set of RefCOCO. AFS: Automatic Feature Selection. SFA: Multi-scale Feature Aggregator. CFS: Coarse-to-Fine Segmentation Decoder. WGC: Word-Level Guided Cross-Attention.

| | Method | val | | | | test A | | | | test B | | | |
|---|---|---|---|---|---|---|---|---|---|---|---|---|---|
| | | P@0.5 | P@0.7 | P@0.9 | IoU | P@0.5 | P@0.7 | P@0.9 | IoU | P@0.5 | P@0.7 | P@0.9 | IoU |
| (a) | baseline | 76.98 | 62.80 | 14.92 | 66.70 | 81.40 | 69.08 | 15.53 | 70.02 | 70.10 | 53.89 | 17.21 | 62.43 |
| (b) | (a)+manual assign | 78.86 | 66.43 | 21.26 | 69.38 | 83.71 | 73.28 | 21.97 | 72.87 | 73.30 | 58.82 | 22.72 | 65.88 |
| (c) | (a)+AFS | 82.32 | 71.77 | 23.95 | 71.78 | 86.3 | 77.35 | 24.02 | 74.79 | 75.89 | 63.49 | 25.84 | 67.58 |
| (d) | (a)+AFS+SFA | 82.37 | 72.34 | 27.09 | 72.39 | 86.30 | 77.44 | 27.68 | 75.25 | 76.52 | 63.47 | 27.36 | 67.97 |
| (e) | (d)+single-branch | 85.97 | 76.76 | 28.72 | 74.65 | 88.45 | 81.36 | 28.76 | 76.54 | 79.50 | 68.45 | 29.55 | 70.62 |
| **(f)** | **(d)+dual-branch(full)** | **86.37** | **77.74** | **29.80** | **75.06** | **89.28** | **82.23** | **29.96** | **77.14** | **80.76** | **69.61** | **30.14** | **71.36** |

that our ASDA, equipped with the CLIP-ViT-B visual backbone, more effectively harnesses the spatial awareness capabilities of the ViT. (2) Compared to ETRIS [54] which uses CLIP ViT-B backbone, ASDA significantly surpasses them by 4.3%, 6.21% and 6.04% in terms of average IoU on RefCOCO, RefCOCO+ and G-Ref datasets respectively. This highlights ASDA's more effective use of CLIP's aligned vision-language features.

## 4.4 Ablation Study

To verify the effectiveness of our proposed components, we conduct comprehensive ablation studies to investigate each component on the RefCOCO val, test A, test B dataset. The components studied include Adaptive Feature Selection (AFS), Scale-Wise Feature Aggregator (SFA), Coarse-to-Fine Segmentation Decoder (CFS), and Word-Guided Coefficient Generator (WCG). The main results of the ablation study are presented in Table 2. Additionally, we have conducted ablation studies on the number of layers $n$ in the Local Visual Attention module, the number of channels $N_C$ in $f_{coef} \in \mathbb{R}^{N_c}$ and $M' \in \mathbb{R}^{HW \times N_c}$ and the architecture of adaptive selection (AS) network, which are presented in Table 3, 4 and 5 respectively. We limit the training of experiments in Table 3, 4, 5 to only 10 epochs and present the result on val split of RefCOCO , which leads to some differences in results compared to those in Table 2.

**Baseline and manual assignment.** (a) The baseline method extracts the highest-level visual feature $F_H$ from the last layer of the visual encoder and fuses it with the global textual feature $f_E$ using element-wise multiplication. The fused feature is then processed by the segmentation decoder to produce the final mask. (b) We enhance the baseline by employing a manual assignment approach following CRIS [51], manually selecting features from Layer 6 and Layer 9 as the low-level feature $F_L$ and middle-level feature $F_M$ respectively. These features $F_L$, $F_M$ and $F_H$ are then fused using the typical concatenation and projection method [51], without the use of multi-scale transformations. This improves 2.68%, 2.85%, 3.45% IoU on RefCOCO val, test A, test B respectively over (a), which demonstrates the significance of utilizing features from intermediate layers which focus on local regions.

**Effect of Adaptive Feature Selection (AFS).** (c) We replace the manual assignment (in Table 2 row 2) with our proposed Adaptive Feature Selection (AFS) module (in Table 2 row 3), which dynamically selects visual features using the global language feature $f_E$. In our setting, AFS will produce muti-scale visual features, here we just use the same fusion and decoder module as in (b). The 2.4%, 1.92% and 1.7% improvement of IoU on three splits compared to (b) shows our AFS module can adaptively choose features that

**Table 3:** Ablation study on the hyperparameter $n$ of the layers in the Local Visual Attention module.

| n | P@0.5 | P@0.7 | P@0.9 | IoU |
|---|---|---|---|---|
| 1 | 82.38 | 71.64 | 23.97 | 71.72 |
| **2** | **83.68** | **73.89** | **24.94** | **72.57** |
| 3 | 82.08 | 71.94 | 25.92 | 72.13 |
| 4 | 83 | 72.37 | 24.79 | 72.57 |

**Table 4:** Ablation study on the hyperparameter $N_C$ which represents the number of channels in $f_{coef} \in \mathbb{R}^{N_c}$ and $M' \in \mathbb{R}^{HW \times N_c}$.

| $N_C$ | P@0.5 | P@0.7 | P@0.9 | IoU |
|---|---|---|---|---|
| 16 | 79.68 | 68.16 | 22.2 | 70.61 |
| **32** | **83.68** | **73.89** | **24.94** | **72.57** |
| 48 | 82.46 | 71.9 | 24.91 | 71.92 |
| 64 | 82.25 | 70.6 | 24.25 | 71.78 |

**Table 5:** Ablation study on the architecture of adaptive selection network (as)

| as | P@0.5 | P@0.7 | P@0.9 | IoU |
|---|---|---|---|---|
| MLP | 81.87 | 71.25 | 23.76 | 71.59 |
| Conv | 82.67 | 72.16 | 24.88 | 72.05 |
| **Linear** | **83.68** | **73.89** | **24.94** | **72.57** |

are more relevant to object described in the referring expression, thereby enhancing segmentation accuracy.

**Effect of Scale-Wise Feature Aggregator (SFA).** (d) We further validate the necessity of the Scale-Wise Feature Aggregator (SFA) module (in Table 2 row 4). Incorporating (c) with the SFA module, the IoU improves by 0.61%, 0.46% and 0.39% on three splits. This highlights the importance of multi-scale visual features in accurately capturing detailed characteristics of objects described in text.

**Effect of Word Guided Dual-Branch Aligner (WGDA).** We compare our proposed Word Guided Dual-Branch Aligner (WGDA) with previously proposed single-branch approaches [7, 17, 51]. As shown in Figure 3, (d) a single-branch approach directly interacts $F_{coarse}$ from the ASFA module with word-level language features $f_T$, while (f) our WGDA method uses dual-branch to let inguistic descriptors directly interact with masks predictions. The results indicate that our WGDA outperforms the previous single-branch method by 0.41%, 0.6%, and 0.74% on three respective splits (in Table 2 row 5, 6). This indicates that our proposed Word Guided Dual-Branch Aligner (WGDA) can not only progressively refine visual features, but also effectively employ word-level language

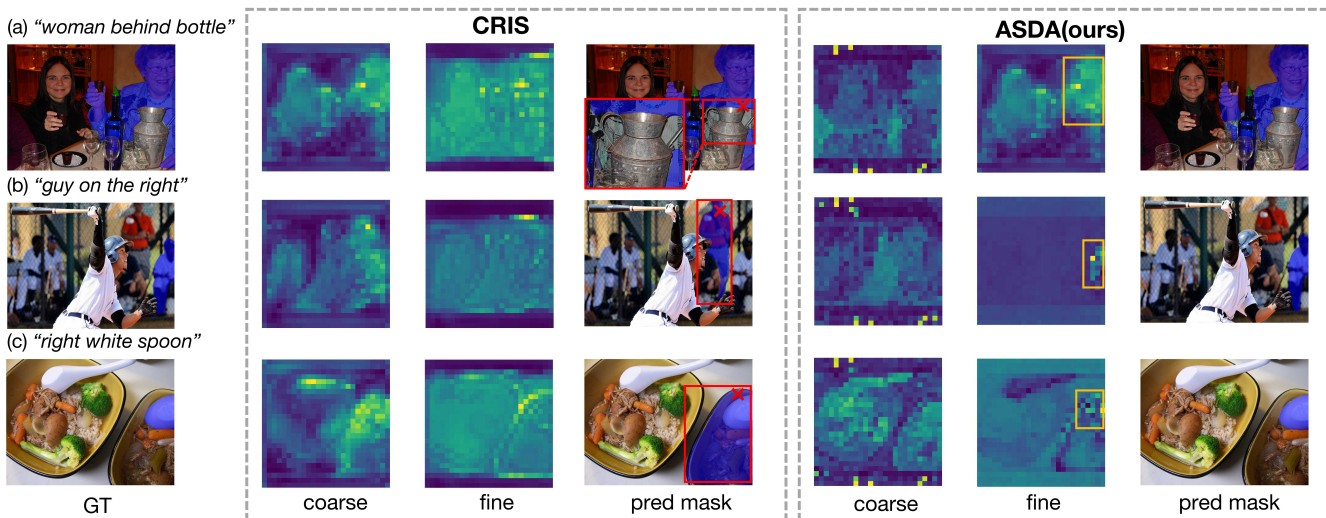

**Figure 4:** Visualization results of feature maps and final predicted mask in CRIS [51] and ASDA respectively. The coarse feature map represents the features without interaction with word-level language features, while the fine feature map results from the fusion and interaction between word-level features afterwards.

features $f_T$ to guide coarse feature $F_{coarse}$ to focus more on word-level language features, highlighting the importance of precise vision-language alignment.

**Abalation about Adaptive selection (AS) network.** In Table 5, we experiment different architectures within the adaptive selection (AS) network. In this experiment, we use our full model, only replacing the AS network layer. Specifically, we investigate three structures in the Adaptive Feature Selection (AFS) module to compute relevance scores: Linear, Convolution (Conv), and Multilayer Perceptron (MLP) layers. Each structure includes a softmax layer that transforms the scores into a probability distribution. Our findings indicate that the simple Linear layer configuration outperformed both the MLP and Conv layers, as evidenced by improvements in IoU of 0.98% and 0.52% over the val split of RefCOCO. This demonstrates that the Linear layer's simplicity and efficiency contribute to better generalization and performance.

**Abalation about hyperparameter** In Table 3, We conduct experiments to determine the most appropriate number of layers for Local Visual Attention within the Coarse-to-Fine Segmentation Decoder (CFS). We find that optimal performance is achieved when $n = 2$. In Equation 10, we obtain the final mask through weight and sum between response mask $M' \in \mathbb{R}^{HW \times N_c}$ and coefficients $f_{coef} \in \mathbb{R}^{N_c}$. As shown in Table 4, we conducted ablation studies on the hyperparameter $N_C$ and found that the best performance is achieved when $N_C = 32$. As $N_C$ exceeds the optimal value 32, performance declines due to overly complex model structure.

### 4.5 Qualitative Analysis

In Figure 4, we visualize the feature maps and the final predicted mask in CRIS [51] and our ASDA.

**Comparison of Feature Maps.** Comparing the coarse feature map, which is the fusion results between visual features from visual encoder and sentence-level language feature $f_E$, both CRIS and ASDA roughly capture the objects' semantic information. However, after interacting with word-level language features, the fine feature

map in CRIS struggles to capture text-relevant features, leading to incorrect predictions. In contrast, the fine feature map from our ASDA, obtained after the Word Guided Cross-attention module in the Word-Guided Coefficient Generator (WCG) branch, can capture local visual detail. As shown by the yellow box in the Figure 4, our method's fine feature map can directly focus on visual features relevant to the text.

**Comparison of Predicted Mask.** The final predicted mask clearly demonstrates the superiority of our method. As indicated by the red box in the figure, for expression (a), CRIS fails to finely segment the part of the woman obscured by a bottle, whereas our method produces a detailed mask. In scenario (b), among multiple easily confusable objects, CRIS struggles to identify the correct person, while ASDA accurately locates the target, as already detected by fine feature map . For (c), CRIS incorrectly identifies the object, whereas ASDA produces an accurate and refined mask. More visualization results are provided in the Appendix.

## 5 CONCLUSION

In this paper, we introduce the Adaptive Selection with Dual Alignment (ASDA) framework for Referring Image Segmentation. Initially, our Adaptive Feature Selection and Fusion (AFSF) module dynamically selects visual features from vision encoder layers related to various descriptive texts. AFSF includes a scale-wise feature aggregator to provide hierarchically aggregated features that preserve crucial low-level details and robust features for dual alignment. Secondly, we utilize a Word Guided Dual-Branch Aligner (WGDA) that integrates visual features with linguistic cues through word-guided attention, effectively addressing vision-language misalignment by allowing linguistic descriptors to directly interact with mask predictions. This ensures focus on relevant image regions for robust predictions. Our extensive experiments show that ASDA outperforms state-of-the-art methods on the RefCOCO, RefCOCO+, and G-Ref benchmarks.

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
