# OpenReview forum: "Adaptive Selection based Referring Image Segmentation"
_acmmm.org/ACMMM/2024/Conference — MM2024 Poster_

### Official Review · Reviewer_YXDK · 2024-05-06

**Rating:** 3
**Confidence:** 3

**Summary:**

This paper proposed a method, ASDA, for refer segmentation. First, an Adaptive Feature Selection and Fusion (AFSF) module is designed to dynamically select visual features to generate coarse features. Then, a Word Guided Dual-Branch Aligner (WGDA) module is leveraged to integrate coarse features with linguistic cues by word-guided attention, producing a word-guided coefficient. Finally, the mask is generated by production of both branch.

**Strengths:**

1. The proposed method demonstrates effectiveness on three employed datasets.
2. The idea of coarse-to-fine is quite novel.

**Limitations:**

1. Why choose vision features on intermediate layers to generate coarse features? Why choose from layer [4,5,6] and [7,8,9] respectively? How the attention region shifts between intermediate layers should be more explained.
2. Why first downsample F_L and F_M, and then upsample them to generate the coarse features?
3. The stage that refines the F_coarse is drawn to be on the text branch, but in fact, it refers to refining the coarse features of the vision branch, instead of injecting vision features into text features. Therefore, the proposed method actually only involves single-branch, that is injecting text features into vision features. It seems more like a dual-stage method, instead of a dual-branch method.
4. Lacks ablation study on layers chosen to generate coarse features ( in AFS ).
5. The concept of "stage" and "branch" should be unified, or it will result in confusion.

**Suitability:**

3

---

### Official Review · Reviewer_ThCY · 2024-05-15

**Rating:** 4
**Confidence:** 2

**Summary:**

This paper proposes a novel framework to solve the RIS task, using text features to guide segmentation for better visual language alignment.  Extensive comparative experiments demonstrate the effectiveness of ASDA and validation experiments demonstrate the effectiveness of the proposed module.

**Strengths:**

1. The WGDA module is a novel design
2. The AFSF uses text features to guide visual feature seletction that can perform more visual-language alignment.
3. ASDF demonstrates strong performance

**Limitations:**

1. Is θas really work？ It only has three neurons
2. It would be more convincing if you could provide the specific design of downsampling  and verify whether it is necessary.
3. It is difficult to understand the specific structure of WCG and CFS based on text descriptions only.

**Suitability:**

3

---

### Official Review · Reviewer_XGpB · 2024-05-25

**Rating:** 3
**Confidence:** 4

**Summary:**

The paper proposes a adaptive selection based method for referring image segmentation. The method adaptively selects features from certain layers from the visual encoder, instead of only using the last layer of the backbone. The paper also propose a scale-wise feature aggregator that fuses the global language features and a Coarse-to-Fine Segmentation Decoder to generate the final mask.

**Strengths:**

The paper is well written and easy to follow.

The framework achieves sota performance on multiple datasets.

**Limitations:**

The architecture of the network looks quite complex, but we have seen many of the designs in previous works. e.g. the scale-wise is actually very similar with the FPN used in MCN, which also involves feature fusion in this step; coarse-to-fine segmentation decoder is quite like the mask decoder in VLT, and word-guided coefficient generator looks like some kind of weighted sum of candidate masks, which is essentially the same with the pipeline used in GRES.

The motivation is established but not well proved. Authors claim that the selected feature of AFS are “language-perferred”, but no evidence are given and more in-depth discussion and analysis are missing. Will certain words affect the selection? What are the difference between features from Layer 4 or Layer 6?

Visualizations of intermediate results are also missing. E.g. all candidate masks M’ and corresponding scores.

Eq.10. the element-wise multiplication of N_c and HW*N_c should also be HW*N_c. Did you mean matrix multiplication?

I am a little confusing with Figure 4: where are these features extracted and how to normalize?

The network also looks very big, better provide the parameter size and FLOPs.


Summary: The main issue is that most of the parts of the network are similar with previous works, and the effectiveness of the proposed adaptive selection mechanism lack solid support.

**Suitability:**

3

---

### Official Review · Reviewer_Dru3 · 2024-06-02

**Rating:** 3
**Confidence:** 3

**Summary:**

The paper proposes a framework for the Referring image segmentation task. Specifically, they propose a method to adaptively select visual features from different layers, instead of using manually selected layers.

The proposed Adaptive Selection with Dual Alignment (ASDA) framework increases performance on Referring image segmentation (RIS) tasks. ASDA contains the Adaptive Feature Selection and Fusion (AFSF) module and the Word Guided DualBranch Aligner (WGDA) module. The AFSF module is composed of Adaptive Feature Selection (AFS) and Scale-Wise Feature Aggregator (SFA) modules. The WGDA module is divided into two branches: the Coarse-to-Fine Segmentation Decoder (CFS) and the Word Guided Coefficient Generator (WCG).

+ The AFSF module dynamically chooses the most relevant visual features from language features, shifting from the traditional fixed-layer selection to a dynamic, text-responsive feature selection process.

+ The WGDA module enhances the alignment and robustness of word-level features with visual features using a dual-branch structure, which improves the interaction between linguistic descriptors and mask predictions.

**Strengths:**

The paper addresses the problem of effectively utilizing vision features from different layers of an encoder, which is a problem worth investigating. The writing is easy to follow.

The proposed method is described clearly, and the framework is illustrated in a comprehensive way. In the experiment, the authors use ViT-B as the visual encoder and Transformer as the language encoder and initialize them with CLIP. The framework achieves better results on all three datasets RefCOCO, RefCOCO+, and G-Ref. This shows that ASDA not only enhances the understanding of the location and appearance information in RefCOCO but also adapts effectively to the diverse expressions in RefCOCO+ and G-Ref.

**Limitations:**

While the idea makes sense, besides standard benchmarks, there are few theoretical arguments or experimental results supporting adaptively selecting the layers over using fixed layers.

The implementation uses a different backbone than previous approaches, making it difficult to compare, especially when the improvement is a few percentage points.

Figure 2 is too detailed. Including the specific architecture of each block is unnecessary and hinders comprehension.

**Suitability:**

2

---

### Meta-Review · Area_Chair_2eWD · 2024-07-03

**Recommendation:** Accept (Poster)
**Confidence:** 4

**Metareview:**

This is a borderline case. This paper introduces an adaptive vision-language features alignment method for referring image segmentation. The original reviews are one borderline accept and three borderline reject. The main concerns center around the complexity of the proposed method and unfair experimental settings. The rebuttal successfully addressed some of the concerns and one negative reviewer raises the final rating to borderline accept. The authors are encouraged to incorporate all the comments in the final version.